# Vemurafenib Inhibits Enterovirus A71 Genome Replication and Virus Assembly

**DOI:** 10.3390/ph15091067

**Published:** 2022-08-27

**Authors:** Bodan Hu, Kenn Ka-Heng Chik, Jasper Fuk-Woo Chan, Jian-Piao Cai, Hehe Cao, Jessica Oi-Ling Tsang, Zijiao Zou, Yin-Po Hung, Kaiming Tang, Lilong Jia, Cuiting Luo, Feifei Yin, Zi-Wei Ye, Hin Chu, Man-Lung Yeung, Shuofeng Yuan

**Affiliations:** 1State Key Laboratory of Emerging Infectious Diseases, Carol Yu Centre for Infection, Department of Microbiology, School of Clinical Medicine, Li Ka Shing Faculty of Medicine, The University of Hong Kong, Pokfulam, Hong Kong SAR, China; 2Centre for Virology, Vaccinology and Therapeutics, Hong Kong Science and Technology Park, Hong Kong SAR, China; 3Hainan Medical University-The University of Hong Kong Joint Laboratory of Tropical Infectious Diseases, Hainan Medical University, Haikou 571199, China; 4Key Laboratory of Tropical Translational Medicine of Ministry of Education, Hainan Medical University, Haikou 571199, China; 5School of Biomedical Sciences, Li Ka Shing Faculty of Medicine, The University of Hong Kong, Pokfulam, Hong Kong SAR, China

**Keywords:** enterovirus, vemurafenib, RAF, MAPK signaling pathway, genome replication, virus assembly, VP0 cleavage

## Abstract

Enterovirus A71 (EV-A71) infection is a major cause of hand, foot, and mouth disease (HFMD), which may be occasionally associated with severe neurological complications. There is currently a lack of treatment options for EV-A71 infection. The Raf-MEK-ERK signaling pathway, in addition to its critical importance in the regulation of cell growth, differentiation, and survival, has been shown to be essential for virus replication. In this study, we investigated the anti-EV-A71 activity of vemurafenib, a clinically approved B-Raf inhibitor used in the treatment of late-stage melanoma. Vemurafenib exhibits potent anti-EV-A71 effect in cytopathic effect inhibition and viral load reduction assays, with half maximal effective concentration (EC_50_) at nanomolar concentrations. Mechanistically, vemurafenib interrupts both EV-A71 genome replication and assembly. These findings expand the list of potential antiviral candidates of anti-EV-A71 therapeutics.

## 1. Introduction

Enterovirus A71 (EV-A71) infection is a major cause of hand, foot, and mouth disease (HFMD), which can be associated with life-threatening neurological complications, such as encephalitis, meningitis, and poliomyelitis-like syndrome [1]. Since its first isolation in 1969 in California, EV-A71 has been recognized as a serious health threat, especially to infants and young children, and has caused major outbreaks in the Asia-Pacific region [1,2]. Despite its clinical importance, there is a lack of treatment options for EV-A71 infection [3].

EV-A71 belongs to the genus Enterovirus of the *Picornaviridae* family. The genus contains various important human pathogens, such as coxsackieviruses, rhinoviruses, echoviruses, and numbered enteroviruses [4]. Enteroviruses are non-enveloped viruses with a single-stranded positive-sensed RNA genome surrounded by the capsid composed of VP0-VP4 [1]. The replication cycle of enteroviruses starts with binding to their receptors, such as scavenger receptor class B member 2 (SCARB2), human tryptophanyl-tRNA synthetase (hWARS), and P-selectin glycoprotein ligand 1 (PSGL1) [4,5]. After entering the host cells through endocytosis, the virions undergo the uncoating process with deformation of capsid proteins and release viral RNA genome into cytoplasm [6]. The genomic replication and transcription take place in virus-induced membrane replication organelles [7]. Following the synthesis of viral proteins and proteolytic processing by viral protease 3CD^pro^, assembly of progeny virions occurs through self-oligomerization of capsid proteins VP0, VP1, and VP3 into protomers and pentamers [8,9]. Incorporation of nascent genome RNA into capsid induces the cleavage of VP0 into VP2 and VP4, generating mature and infectious progeny virions [10,11].

Viruses can manipulate host machinery to regulate host antiviral responses and facilitate viral replication. The RAF-MEK-ERK pathway, one of the three mitogen-activated protein kinases (MAPK) cascades, plays an important role in the regulation of cell proliferation, differentiation, and apoptosis [12]. The kinase signaling pathway is induced by extracellular agents including both DNA and RNA viruses [13]. Aberrant regulation takes place during infection of various viruses, such as influenza virus, hepatitis C virus, coxsackievirus, Ebola virus, and coronaviruses [13,14,15,16]. Inhibition of the MAPK/ERK pathway leads to retention of viral ribonucleoproteins (RNPs) of influenza viruses in the cell nucleus and impairs viral replication [14].

Enteroviruses such coxsackievirus B3 and EV-A71 induce a biphasic activation of the MAPK/ERK signaling pathway immediately after virus binding to receptors and in the late stage of infection [16,17]. ERK pathway inhibitors and siRNA against ERK inhibit EV-A71 infection [18]. EV-A71-induced ERK signaling activation may promote cyclooxygenase-2 expression, one of the factors contributing to neurological inflammation during infection [19]. Therefore, targeting the RAF-MEK-ERK pathway may be a useful therapeutic strategy for EV-A71 infection.

Drug repositioning has been increasingly applied to identify potential therapeutics for viral infections [20,21]. It has several potential advantages over de novo drug development, such as significant reduction in cost and time due to known safety and pharmacokinetics data, and an increase in drug approval rates. Therefore, we employed the drug repositioning strategy to investigate whether vemurafenib, an FDA-approved RAF inhibitor for treating BRAF^V600^ mutant-related melanoma, could be used for the treatment of EV-A71 infection. Vemurafenib selectively inhibits the activating BRAF^V600^ mutant kinase to block the downstream BRAF-MEK-ERK signaling transduction. In turn, it reduces aberrant melanoma cell proliferation, promoting cell apoptosis [22,23]. We showed that vemurafenib potently inhibits EV-A71 in vitro and activates the MAPK/ERK kinase cascades and restricts virus genome replication and virus assembly.

## 2. Results

### 2.1. Vemurafenib Potently Inhibits EV-A71 Infection

To investigate whether vemurafenib possesses an antiviral activity against EV-A71, we first conducted a cytopathic effect (CPE) inhibition assay to assess vemurafenib’s ability to inhibit virus-induced CPE formation in human rhabdomyosarcoma (RD) cells. As shown in Figure 1A,B, vemurafenib dose-dependently inhibited EV-A71-induced CPE in RD cells, with half maximal effective concentration (EC_50_) and 50% cytotoxicity concentration (CC_50_) of 0.34 ± 0.01 µM and 17.71 ± 0.58 µM, respectively. In the viral load reduction assay, vemurafenib also demonstrated a dose-dependent inhibitory effect against EV-A71 (Figure 1C). The antiviral activity of vemurafenib was similar to that of the broad-spectrum viral polymerase inhibitor remdesivir, which is known to be active against EV-A71, SARS-CoV-2, and other RNA viruses [24,25,26]. Importantly, the selectivity index of vemurafenib (52.1) was even higher than that of remdesivir (35.5), indicating the potential of vemurafenib as a repositioned drug for EV-A71 infection.

### 2.2. Time-of-Drug-Addition of Vemurafenib on EV-A71 Infection

To explore which stage(s) of EV-A71 life cycle is targeted by vemurafenib, we performed a time-of-drug-addition assay [27], in which RD cells were treated with vemurafenib at different time points before, concurrently with, or after virus infection (Figure 2A). One cycle of EV-A71 replication is usually finished within 6–8 h [28,29]; therefore, we collected samples for analysis 10 h after infecting cells at a high multiplicity of infection (MOI = 1). There was a significant reduction in titers of progeny virions in the culture supernatant when RD cells were subjected to treatment at either before (−1 h), concurrently with (0 h), or at latest 4 h after infection, and the reduction was more prominent with earlier treatment (Figure 2B). Vemurafenib exhibited no inhibitory effect when the infected cells were treated at 6 or 8 h post-infection (hpi). To further characterize the antiviral activity of vemurafenib, we analyzed viral RNA synthesis and protein translation in cells. As shown in Figure 2C, cellular viral load also decreased with drug addition at earlier time points. The inhibition was evident even when vemurafenib was added at 6 hpi or 8 hpi, and a similar pattern of gradual decrease in antiviral activity was noted. Indirect immunofluorescence assay showed that EV-A71 VP1 protein expression was also affected (Figure 2D). Altogether, these results suggested that vemurafenib does not function as an entry inhibitor or target viral functional proteins such as protease or polymerase. Instead, it likely induces changes in host cells to impair viral genome replication and transcription, resulting in reduced viral protein translation.

### 2.3. Virus Assembly Was Impaired with Vemurafenib Treatment

At the later stages of EV-A71 life cycle, viral structural capsid proteins, VP0, VP1, and VP3, assemble into heterotrimeric protomers and then pentamers to initiate assembly of progeny virions [8,10,11]. The subsequent VP0 cleavage into VP2 and VP4 is a critical step to generate fully mature virions [8]. To investigate whether the assembly of EV-A71 is affected by vemurafenib, we analyzed the amount of VP2 relative to VP0 when infection was performed in the presence of vemurafenib at a concentration that did not completely inhibit virus replication. As shown in Figure 3A, at low MOI of 0.01, virions produced under the pressure of vemurafenib have less VP2 than those treated with DMSO. The successfully cleaved product VP2 in the virions produced in the presence of vemurafenib was only ~50% of those treated with DMSO (Figure 3B). The defect is even more obvious when infection was performed at a high MOI of 1, in which it takes shorter time until all cells die of infection. A very faint band corresponding to VP2 was observed, whereas VP1 and VP0 were easily detected (Figure 3C). Taken together, these results indicated that the VP0 cleavage during virus assembly is disturbed when vemurafenib is added as an antiviral drug.

### 2.4. RAF-MEK-ERK Signaling Pathway Was Activated by Vemurafenib

Vemurafenib is used for treatment of BRAF^V600^ mutant metastatic melanoma. It highly selectively inhibits BRAF^V600^ mutation-induced activation of the RAF-MEK-ERK signaling pathway, thus stopping cell proliferation and promoting apoptosis of BRAF^V600^ melanoma cells. Mutant BRAF inhibitors have been shown to activate this MAPK kinase cascade in wild-type BRAF cells [30,31], which is also activated by EV-A71 infection [17,18]. Therefore, we investigated the RAF-MEK-ERK kinase responses to vemurafenib in RD cells alone or together with EV-A71 infection. Western blot showed that under normal conditions without virus infection, vemurafenib induces activation of the RAF-MEK-ERK kinase cascade as evident by increased phosphorylated MEK1/2 and ERK1/2 (Figure 4). The activation started as early as 2 h after treatment and persisted for at least 8 h. We also compared the effect of vemurafenib on cells with or without EV-A71 infection. Surprisingly, similar levels of phosphorylated MEK and ERK were detected (lane 2 vs lane 4 at each time point), indicating that EV-A71 had no additional effect on inducing MAPK/ERK kinase cascade at these time points. We additionally analyzed the activation of MEK-ERK kinase signaling at earlier time points, i.e., 15 min, 30 min, and 60 min, p.i., of EV-A71 infection. The results in Appendix A showed that vemurafenib treatment caused general but less significant increase in phosphorylation of MEK1/2 and ERK1/2, regardless of EV-A71 infection or not. These observations indicate that vemurafenib induces a rapid activation of RAF/MEK/ERK kinase cascade, and the effect is more prominent with long-time treatment. Altogether, the intrinsic property of vemurafenib to activate the RAF/MEK/ERK kinase cascade can potentially interfere with later stages of EV-A71 replication.

## 3. Discussion

In this study, we repositioned the clinically available vemurafenib as a potential anti-EV-A71 therapeutic. Our results showed that vemurafenib potently inhibits EV-A71 in vitro, with an EC_50_ value in the nanomolar range. Previous studies showed that vemurafenib limits influenza A virus replication in A549 and calu-3 cells and inhibits human echovirus 1 infection in A549 cells, suggesting that vemurafenib or targeting MAPK kinase cascade could be a broad-spectrum antiviral strategy [32,33]. Vemurafenib is a drug approved for treatment of both adult and pediatric patients with unresectable or metastatic melanoma with the BRAF^V600^ mutation [22,34,35]. Despite that the drug functions as a competitive inhibitor to block activation of the RAF-MEK-ERK signaling pathway in cells with mutant BRAF, we found that this inhibitor constantly activates the RAF signaling in RD cells which have a wild-type BRAF. The activation is more prominent with long-time treatment, and it is negligible at early time points of EV-A71 infection. The hyperactivation of RAF-MEK-ERK kinase cascade by vemurafenib has also been reported in a study on influenza A virus infection [33]. Vemurafenib activates CRAF in wild-type BRAF cells and the subsequent MEK/ERK phosphorylation [23,30,31]. Vemurafenib inhibits the activation of influenza A virus- or tumor necrosis factor alpha (TNFα)-induced JNK and p38 kinase signaling cascades, and the other two MPAK signaling pathways [12,33]. EV-A71 infection leads to activation of Jun-N-terminal kinase 1/2 (JNK1/2) [36] and induction of proinflammatory cytokines regulated by the MAPK p38 signaling cascades [37], thus regulating cell apoptosis and inflammation. Therefore, vemurafenib probably also limits the activation of the two MAPK signaling cascades induced by EV-A71, which may contribute to its inhibition of EV-A71 infection.

Given the complexity of the MAPK pathway and the fact that there are over 100 proteins involved in the downstream of ERK activation [12], activated ERKs induced by vemurafenib may target downstream proteins that are different from those induced by EV-A71 infection. In normal cells, activated ERKs are translocated to the cell nucleus and transactivate transcription factors to stimulate genes to promote cell mitosis and growth [12]. Furthermore, inhibition of RAF-MEK-ERK signaling impairs nuclear export of viral ribonucleoprotein complexes (RNPs) of influenza A viruses which exploits cell nucleus for genome replication [14]. However, EV-A71 replication happens in a virus-induced unique replication organelle instead of cell nucleus [7]. The ROs formation is aided by host proteins including phosphatidylinositol 4-kinase-β (PI4KB) and oxysterol-binding protein (OSBP) [4,38]. Thus, the strategy of EV-A71 used for genome replication should be different from that used by influenza A viruses or host cell proliferation. On the other hand, EV-A71 infection-induced MAPK activation has been shown to cross talk with TNF-α-related cell apoptosis with the TNF-α induction appearing to be suppressed by ERK signaling to inhibit extrinsic apoptosis in EV-A71-infected cells [37]. The activated ERK signaling also interferes with the NF-kB pathway to upregulate expression of proinflammatory cytokines IL-6 and IL-8 and, more importantly, to increase viral replication [37]. Our results revealed that EV-A71 genome replication is impaired by vemurafenib treatment, and earlier addition of inhibitor leads to more severe defect. Altogether, it would support the hypothesis that vemurafenib competes with EV-A71 to activate MAPK/ERK pathway to promote gene expression for cell growth in the nucleus, which fails to provide support, e.g., PI4KB, for EV-A71 ROs formation and genome replication. The detailed involvement of the MAPK signaling pathway with ROs formation should be further investigated. 

A crucial step in virions maturation is VP0 cleavage into VP2 and VP4. We also found that vemurafenib treatment led to reduced VP2 production, indicating a defect in virus assembly. The VP0 cleavage is potentially mediated by incorporation of viral genome RNA into capsid [4,10]. Therefore, the impaired viral genome replication caused by vemurafenib might be responsible for the affected VP0 cleavage. However, the exact mechanism underlying VP0 cleavage remains poorly understood. Other host factors such as sterile alpha motif and histidine-aspartic acid domain-containing protein 1 (SAMHD1) and heat shock protein-90-beta (Hsp90β) are also involved in this process [29,39]. A link between MAPK signaling and SAMHD1 or Hsp90β requires further exploration.

In conclusion, vemurafenib demonstrates antiviral activity against EV-A71 infection. It induces RAF-MEK-ERK signaling in wild-type BRAF cells, potentially to promote cell mitosis and proliferation, which paradoxically impairs viral genome replication and virus assembly. Our findings provide new insights for developing antiviral inhibitors targeting the MAPK signaling pathway as a potential broad-spectrum antiviral strategy. Nevertheless, the complexity of the kinase cascade should be carefully examined to restrict potential host toxicity and side effects. 

## 4. Materials and Methods

### 4.1. Cells, Viruses, and Reagents

Human rhabdomyosarcoma (RD, ATCC, CCL-136™) cells were maintained in Dulbecco’s Modified Eagle’s Medium (DMEM, Gibco, Waltham, MA, USA) supplemented with 10% heat-inactivated fetal bovine serum (FBS, Gibco, Waltham, MA, USA), 100 units/mL of penicillin and 1000 µg/mL of streptomycin (Gibco, Waltham, MA, USA) as previously described [40]. Human EV-A71 viruses (isolate MY104-9-SAR-97, GenBank: DQ341368) were cultured in RD cells in serum-free DMEM supplemented with 2% FBS. Virus stocks were stored at −80 °C and were titrated by a 50% tissue culture infection dose (TCID50) assay as previously described [41]. Vemurafenib (PLX4032) and remedesivir (GS-5734) were purchased from MedChemExpress (MCE) and dissolved in dimethylsulfoxide (DMSO) to prepare 20 mM stock.

### 4.2. Virus Inhibition Assay

The EV-A71 CPE inhibition assay was performed as previously described with slight modifications [27]. Briefly, the drug compounds were three-fold serially diluted with serum-free DMEM and added to confluent RD cells in 96-well culture plates (30,000 cells/well) in triplicate for 1 h at 37 °C. After incubation, the drug-containing media were removed, and EV-A71 (MOI of 0.01) was added together with fresh drug-containing media to each well. After adsorption for 1 h at 37 °C, the virus-compound mixture was removed, and the cells were washed with DMEM once to remove unbound virus. Subsequently, media with antiviral compounds were added to the cells for further incubation for 24–48 h at 37 °C in 5% CO_2_. To analyze inhibition of cytopathic effect (CPE), 50 µL of Promega cell-titer glo was added to each well 48h post infection and luminescence was then measured by Promega plate reader. To analyze virus yields in culture supernatant and virus genome replication in cells, culture supernatant (48 hpi) and cells were harvested (24 hpi) for analysis. Viral RNA was extracted using RNeasy Mini Kit (Qiagen, Hilden, Germany) according to the manufacturer’s instruction and subjected to real-time RT-PCR using One Step TB Green^®^ PrimeScript™ RT-PCR Kit II (TaKaRa) with primers targeting 5′-UTR of viral genome (forward primer: 5′- GCCCCTGAATGCGGCTAAT -3′, reverse primer: 5′- ATTGTCACCATAAGCAGCCA -3′). Cellular relative gene expression was normalized to GAPDH internal control (forward primer: 5′-GCCTCTTGTCTCTTAGATTTGGTC-3′, reverse primer: 5′-TAGCACTCACCATGTAGTTGAGGT-3′).

### 4.3. Time-of-Drug-Addition Assay

A Vemurafenib time-of-addition assay was performed as previously described with minor modifications [27]. Briefly, RD cells were seeded into 24-well plates (200,000 cells/well) or 8-well chamber slides (65,000 cells/well) one day before infection with EV-A71 at an MOI of 1. After adsorption for 1h, cells were washed once and changed to fresh media. Vemurafenib (2 µM) was added to the media at different time point (−1 h, 0 h, 2 h, 4 h, 6 h or 8 h) after inoculation. For the time point of “−1 h”, RD cells were preincubated with vemurafenib for 1h before infection, and the compound was kept in the fresh media afterwards. For the time point of “0 h”, vemurafenib was added together with EV-A71 inoculation and kept in the fresh media afterwards. Culture supernatant and cells were harvested at 10 hpi, and viral genome copy numbers were determined. To analyze VP1 expression, cells in the 8-well chamber slides were fixed after infection for 10 h, permeabilized, blocked, and then stained with in-house produced primary anti-VP1 antibody and anti-mice secondary antibody Alexa Fluor 488. Images were taken with Olympus BX53 equipped with an LED illuminator.

### 4.4. Virus Assembly Assay

RD cells in T175 flask were ~90–100% confluency and infected with EV-A71 at an MOI of 0.01 or 1. Vemurafenib (or DMSO) was added to the media after adsorption for 1 h. Culture media was harvested when over 50% cells showed CPE. After clearing cell debris by low-speed centrifugation (4000× *g*, 5 min), viruses were pelleted by ultracentrifugation (175,000× *g*, 1.5 h) through 20% sucrose cushion and resuspended in 1×TNE buffer (10 mM Tris, 100 mM NaCl und 1 mM EDTA, pH 7.4) as previously described [42], and then subjected to western blot analysis.

### 4.5. Western Blot

To analyze virus assembly, pelleted viruses were mixed with loading buffer containing β-mercaptoethanol, separated by SDS-PAGE, and blotted onto polyvinylidene difluoride (PVDF) membranes. In-house mouse monoclonal antibodies targeting VP2 and VP1 were used to detect both VP0 and VP2, and VP1, respectively.

To analyze the activation of MAPK pathway, RD cells were infected without or with EV-A71 at an MOI of 5 and concurrently treated with 4 µM vemurafenib or 0.02% DMSO for 1 h. Cells were then washed once with DMEM and replenished with serum-free DMEM. Cells were collected at varied time points (15 min, 30 min, and 60 min, 2 h, 4 h, 6 h, and 8 h) post-infection and lysed in radioimmunoprecipitation assay buffer (RIPA buffer, ThermoFisher, Waltham, MA, USA) supplemented with protease and phosphatase inhibitors (#A32961, ThermoFisher, Waltham, MA, USA). Proteins in cell lysates were separated by SDS-PAGE and blotted onto PVDF membranes. Phospho-MEK1/MEK2 (Ser217, Ser221) Monoclonal Antibody (#MA5-15016, Invitrogen, Waltham, MA, USA) and Phospho-ERK1/ERK2 (Thr185, Tyr187) Polyclonal Antibody (#44-680G, Invitrogen) were used to detect phosphorylated MEK and ERK. P44/p42 MAPK (ERK 1/2) antibody (cell signaling, #9102) was used to detect endogenous level of total ERK 1/2 protein. Antibodies targeting γ-tubulin was purchased from Sigma (#T6557).

After washing away unbound primary antibodies, secondary antibody coupled with horseradish peroxidase (HRP) were incubated with membranes. Immobilon classico western HRP substrate (Millipore, Burlington, MA, USA) was applied to membrane to give chemiluminescence signals which was then detected by chemiluminescence imaging system (Alliance Q9, Uvitec, St. John’s, NL, Canada). The density of bands was measured by Image J.

### 4.6. Statistical Analysis

Statistical comparisons between different groups were performed by Student’s *t*-test using GraphPad Prism 8. *p* < 0.05 was considered statistically significant.

## Figures and Tables

**Figure 1 pharmaceuticals-15-01067-f001:**
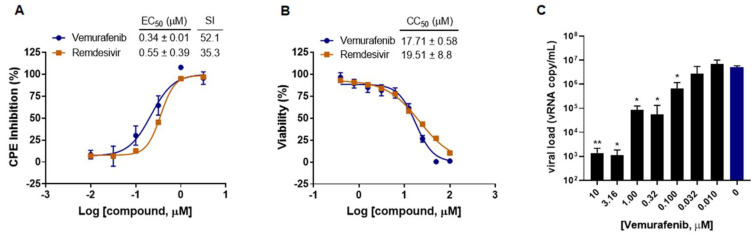
Vemurafenib significantly inhibits EV-A71 infection. (**A**) EV-A71 infection (MOI = 0.01) induced CPE inhibition by and (**B**) cytotoxicity of vemurafenib and remdesivir (as a control) in RD cells. (**C**) Viral load in the culture supernatant at the presence of serially diluted vemurafenib. Data are representative of three independent experiments and displayed as means ± standard deviation. Asterisks (*) indicate statistically significant differences (** *p* < 0.01 and * *p* < 0.05, Student’s *t*-test) between vemurafenib treatment and mock treatment (0 µM, colored in blue). EC_50_ and CC_50_ were obtained with non-linear regression analysis. EC_50_: half maximal effective concentration to completely inhibit EV-A71 infection. CC_50_: 50% cytotoxic concentration. SI: selective index, CC_50_/EC_50_.

**Figure 2 pharmaceuticals-15-01067-f002:**
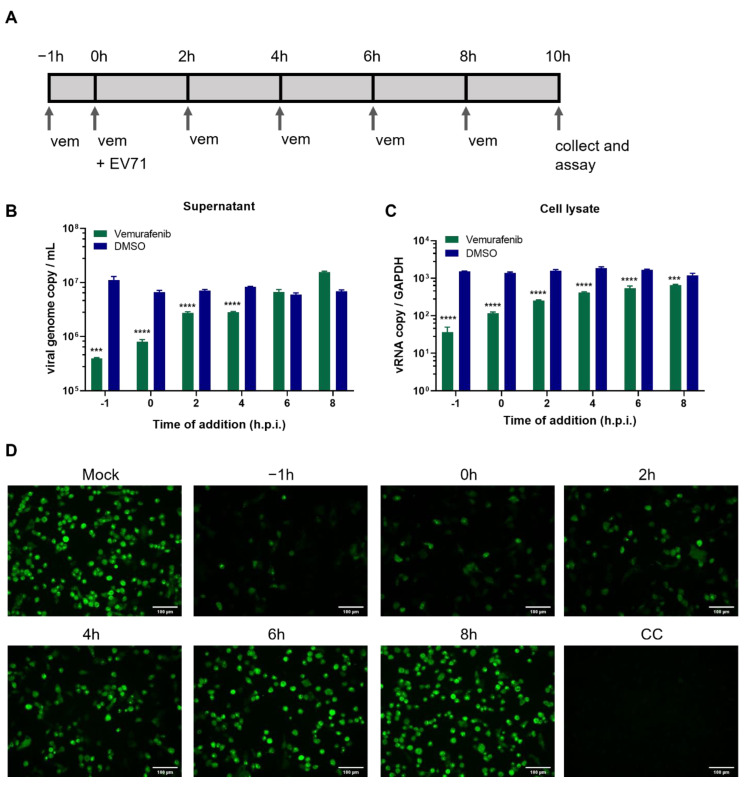
The time-of-drug-addition assay with vemurafenib. RD cells were infected with EV-A71 at an MOI of 1, and treated with 2 µM vemurafenib at varied time points, including pre-infection (−1 h), co-infection (0 h), and post-infection (2 h, 4 h, 6 h, or 8 h). (**A**) Graphical scheme illustrating the experimental design. (**B**) The amount of viral RNA released into culture supernatant, measured by real-time RT-PCR using primer targeting 5′-UTR. (**C**) The relative amount of viral genome RNA in the cells. Data are representative of three independent experiments. Asterisks (*) indicate statistically significant differences (*** *p* < 0.001, **** *p* < 0.0001) between vemurafenib and DMSO according to a Student’s *t*-test. (**D**) VP1 expression detected by indirect immunofluorescence. Mock: mock treatment, RD cells were infected EV-A71 but without vemurafenib treatment. CC: mock infection, RD cells only, without infection nor vemurafenib treatment. Scale bar equals to 100 µm. Representative images are shown.

**Figure 3 pharmaceuticals-15-01067-f003:**
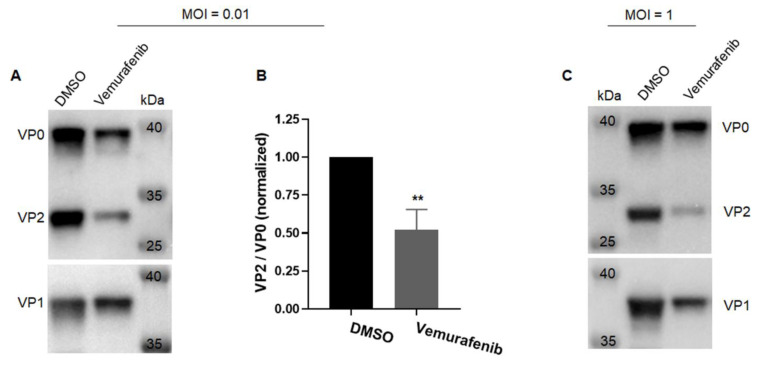
Virus assembly is impaired with vemurafenib treatment. (**A**,**B**) RD cells were infected with EV-A71 at an MOI of 0.01 and concurrently treated with 1 µM vemurafenib or 0.005% DMSO. Viruses in culture supernatant were pelleted through 20% sucrose cushion and subjected to western blot. VP0 and VP2 were detected with anti-VP2 antibody. VP1 was detected using anti-VP1 antibody. (**A**) The amount of VP0, VP2, and VP1 in purified viruses. (**B**) Quantification of relative amount of VP2 to VP0. Density of VP0 and VP2 bands was determined and the VP2/VP0 ratio was calculated and normalized to that treated with DMSO. Results from three virus preparations are displayed as means ± standard deviation. Asterisks (*) indicate statistically significant differences (** *p* < 0.01) between DMSO mock treatment group and vemurafenib-treated group according to a Student’s *t*-test. (**C**) The amount of VP0, VP2, and VP1 in purified viruses. RD cells are infected with EV-A71 at an MOI of 1 and concurrently treated with 2 µM vemurafenib or 0.01% DMSO. One representative result from two different virus preparation is shown.

**Figure 4 pharmaceuticals-15-01067-f004:**
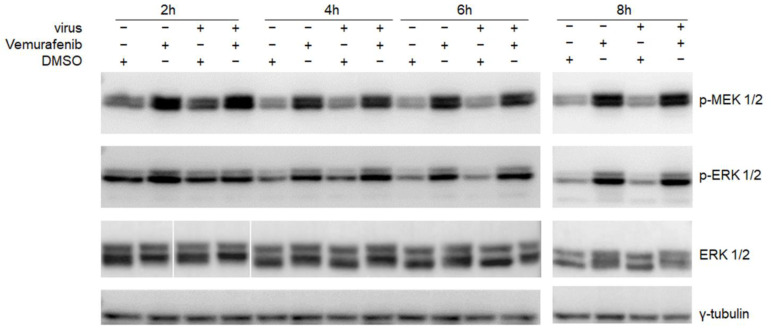
RAF-MEK-ERK kinase cascade was activated by vemurafenib treatment. RD cells were infected without or with EV-A71 at an MOI of 5 and concurrently treated with 4 µM vemurafenib or 0.02% DMSO. Cells were collected at 2 h, 4 h, 6 h, or 8 h post infection. Activation of RAF-MEK-ERK kinase cascade was analyzed by western blot. Phosphorylated MEK1/2 and ERK1/2 were detected using anti-phospho-MEK1/2 (Ser217/221) antibody and anti-phospho-ERK1/2 (Thr185/187) antibody. Unphosphorylated ERK and γ-tubulin were also detected as controls. Blots are representative data of two independent experiments.

## Data Availability

Data is contained within the article and supplementary material.

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
