# Peer review of "Vemurafenib Inhibits Enterovirus A71 Genome Replication and Virus Assembly"

_pharmaceuticals, 2022, doi:10.3390/ph15091067_

Round 1

Reviewer 1 Report

This article presents an interesting and scientifically sound study, publication is recommended after the minor revisions listed below.

Lines 52-60: A few references should be added for the various statements reported.

Line 73: “cyclooxygenase2” should be changed to “cyclooxygenase-2”

Line 81 (and following lines): when vemurafenib is introduced, more details should be given on its mechanism of action against melanoma, and the rationale to evaluate it as a potential anti-EV71 agent should be better clarified. In addition, it should be better clarified that this is the very first time that vemurafenib is evaluated as an antiviral agent. If this is not the case, previous literature must be discussed accordingly.

Some typos/grammar mistakes are present throughout (e.g., line 113: “we performed time-of-drug-addition assay”, missing article, the sentence should read “we performed a time-of-drug-addition assay”; line 154: “than those of treated with DMSO”, the “of” should be removed; lines 157-158: “in which it takes shorter time until all cells died of infection”, wrong tense used and wrong sentence construction, the sentence should read “in which it takes a shorter time for all cells to die of infection”; etc.). These should be carefully addressed before publication.

In the conclusions, potential caveats of using the the MAPK signaling pathway as an antiviral target should be discussed.

Author Response

This article presents an interesting and scientifically sound study, publication is recommended after the minor revisions listed below.

Point 1: Lines 52-60: A few references should be added for the various statements reported.

Response 1: References (6-11) have been added to this part.

Point 2: Line 73: “cyclooxygenase2” should be changed to “cyclooxygenase-2”

Response 2: Thanks. It has been corrected.

Point 3: Line 81 (and following lines): when vemurafenib is introduced, more details should be given on its mechanism of action against melanoma, and the rationale to evaluate it as a potential anti-EV71 agent should be better clarified.

Response 3: We have added the following sentences to the introduction “Therefore, we employed the drug repositioning strategy to investigate whether vemurafenib, an FDA-approved RAF inhibitor for treating BRAFV600 mutant-related mela-noma, could be used for the treatment of EV-A71 infection. Vemurafenib selectively inhibits the activating BRAFV600 mutant kinase to block the downstream BRAF-MEK-ERK signaling transduction. In turn, it reduces aberrant melanoma cell proliferation and promoting cell apoptosis”

Point 4: In addition, it should be better clarified that this is the very first time that vemurafenib is evaluated as an antiviral agent. If this is not the case, previous literature must be discussed accordingly.

Response 4: We have added the following sentences in the discussion: “Previous studies showed that vemurafenib limits influenza A virus replication in A549 and calu-3 cells and inhibits human echovirus 1 infection in A549 cells, suggesting that vemurafenib or targeting MAPK kinase cascade could be a broad-spectrum antiviral strategy.”

Point 5: Some typos/grammar mistakes are present throughout (e.g., line 113: “we performed time-of-drug-addition assay”, missing article, the sentence should read “we performed a time-of-drug-addition assay”; line 154: “than those of treated with DMSO”, the “of” should be removed; lines 157-158: “in which it takes shorter time until all cells died of infection”, wrong tense used and wrong sentence construction, the sentence should read “in which it takes a shorter time for all cells to die of infection”; etc.). These should be carefully addressed before publication.

Response 5: Thanks. We have read through the manuscript carefully and made corrections to those typos/grammar mistakes.

Point 6: In the conclusions, potential caveats of using the MAPK signaling pathway as an antiviral target should be discussed.

Response 6: Thanks for raising up this important comment. We have added the following sentences in the conclusion: “Our findings provide ……… as a potential broad-spectrum antiviral strategy. Nevertheless, the complexity of the kinase cascade should be carefully examined to restrict potential host toxicity and side effects.”

Reviewer 2 Report

Comments on “Vemurafenib inhibits enterovirus A71 genome replication and virus assembly.”

EV-A71 is an RNA virus belonging to the Picornaviridae family. It is responsible for outbreaks of HFMD and is associated with neurological pathologies. Currently, antivirals to treat EV-A71 are severely limited. Here, Bodan Hu et al. describe potent antiviral activity of a widely approved chemotherapeutic compound, vemurafenib, which is used to treat late-stage melanoma by inhibiting a mutant version of B-Raf. The authors demonstrate that vemurafenib treatment of RD cells significantly inhibits EV-A71 replication. Furthermore, they show that vemurafenib treatment attenuates viral replication when added up to 4+ hours after viral infection, suggesting that the mechanism of vemurafenib’s antiviral activity acts late in the viral replication cycle. In addition to attenuating replication of the viral genome, the authors go on to show that vemurafenib treatment compromises the proteolytic processing of viral structural proteins. Mechanistically, the authors demonstrate that vemurafenib treatment potently activates the WT Raf-MEK-ERK signaling cascade, which may have consequences for viral replication.

Comments and concerns:

·       The authors present a relatively thorough characterization of the impact of vemurafenib on EV-A71 replication. They identify two specific aspects of the viral life cycle that appear to be impacted by vemurafenib treatment. This may be sufficiently interesting to the field to warrant publication.

·        The mechanism by which vemurafenib treatment counteracts viral genome replication and virion maturation is rather weak and correlative. It is unclear to me how activation of the Raf-MEK-ERK pathway inhibits EV-A71 replication. Do other modulators of this pathway impact EV-A71 replication?

·        HFMD generally affects young children. Are there safety concerns regarding the treatment of young children with vemurafenib?

·        Total levels of MEK1/2 and ERK1/2 are not shown.

Author Response

  • The authors present a relatively thorough characterization of the impact of vemurafenib on EV-A71 replication. They identify two specific aspects of the viral life cycle that appear to be impacted by vemurafenib treatment. This may be sufficiently interesting to the field to warrant publication.
  • Point 1: The mechanism by which vemurafenib treatment counteracts viral genome replication and virion maturation is rather weak and correlative. It is unclear to me how activation of the Raf-MEK-ERK pathway inhibits EV-A71 replication. Do other modulators of this pathway impact EV-A71 replication?

    Response 1: Thank you very much for raising out this important issue. As mentioned in the second paragraph of discussion, we believe that activation of the Raf-MEK-ERK pathway inhibits EV-A71 replication organelle (RO) formation. Given the complexity of the MAPK pathway and the fact that there are over 100 proteins involved in the downstream of the pathway, further investigations to pinpoint the exact binding molecules are warranted. In this stage, we do not exclude the possibility of other modulators of this pathway to play a role in EV-A71 replication cycle.

Point 2: HFMD generally affects young children. Are there safety concerns regarding the treatment of young children with vemurafenib?

Response 2: Thank you for the comments. We have carefully checked the literature and found that the safety data of vemurafenib treatment in young children were conducted in pediatric patients with BRAFV600 melanoma, whose safety profile is similar to that reported in adults; whereas the safety data of vemurafenib treatment in healthy adults and children without tumor should be re-evaluated. We have added the following words in the discussion: Vemurafenib is a drug approved ……. both adult and pediatric patients ………

Point 3: Total levels of MEK1/2 and ERK1/2 are not shown.

Response 3: Thank you and we have performed additional experiments in Supplementary Figure 1. The total level of ERK1/2 are included in the revised manuscript. The expression level is similar to that of γ-tubulin.

Reviewer 3 Report

Hu and colleagues investigated the possibility of using a targeted chemotherapeutic drug against melanoma (vemurafenib), to treat infection with the A71 enterovirus. Although this is an infection that has the potential to cause severe neurological problems and encephalitis in young children, vemurafenib has some serious side effects upon administration that could make administration of the drug dangerous in most cases and this should be acknowledged in the discussion section.

- The authors have done a good job dissecting the viral life cycle, however I am still not convinced about the implications of their work on RAF-MEK-ERK pathway. ERK1/2 signalling remains sustained upon activation with certain stimuli, however ERK1/2 is known to be phosphorylated pretty quickly too. There is a chance that the authors missed the actual A71-mediated phosphorylation of MEK/ERK. They do mention this at the discussion section, but it would be very interesting to see what occurs at earlier time points, especially with a virus that completes its life cycle within 8h. HCV virus, another small RNA virus is known to have entered the cell within 1h following infection and it induces MAPKs even at the stage of endocytosis, so I cannot see how they expect to observe changes in the pathway from 2h onwards.   Also, why are they not using total MEK/ERK antibodies instead of tubulin, to show steady (or not) expression of the respective proteins?

- Have the authors checked other MAPK pathways that have been involved in A71 replication, such as the JNK pathway. Vemurafenib down-regulates JNK and p38, so it is possible that the observed effect may be due to these cascades at least in part.

Author Response

Response to Reviewer 3 Comments

Point 1: Hu and colleagues investigated the possibility of using a targeted chemotherapeutic drug against melanoma (vemurafenib), to treat infection with the A71 enterovirus. Although this is an infection that has the potential to cause severe neurological problems and encephalitis in young children, vemurafenib has some serious side effects upon administration that could make administration of the drug dangerous in most cases and this should be acknowledged in the discussion section. The authors have done a good job dissecting the viral life cycle, however I am still not convinced about the implications of their work on RAF-MEK-ERK pathway. ERK1/2 signalling remains sustained upon activation with certain stimuli, however ERK1/2 is known to be phosphorylated pretty quickly too. There is a chance that the authors missed the actual A71-mediated phosphorylation of MEK/ERK. They do mention this at the discussion section, but it would be very interesting to see what occurs at earlier time points, especially with a virus that completes its life cycle within 8h. HCV virus, another small RNA virus is known to have entered the cell within 1h following infection and it induces MAPKs even at the stage of endocytosis, so I cannot see how they expect to observe changes in the pathway from 2h onwards. Also, why are they not using total MEK/ERK antibodies instead of tubulin, to show steady (or not) expression of the respective proteins?

Response 1: Thank you very much for these insightful comments. We agree with this reviewer to tone down the clinical potential of vemurafenib against EV-A71 infection in children, while RAF/MEK/ERK as a broad-spectrum antiviral target might be of general interests. To strengthen the mechanistic investigation, we have performed additional experiment to analyze the activation of MEK-ERK kinase signaling at earlier time points and using total ERK antibodies (please refer to new figure S1 for detail). We have also added following interpretation in the results: “The results in figure S1 showed that vemurafenib treatment caused general but less significant increase in phosphorylation of MEK1/2 and ERK1/2, regardless of EV-A71 infection or not. These observations indicate that vemurafenib induces a rapid activation of RAF/MEK/ERK kinase cascade, and the effect is more prominent with long-time treatment.”

Point 2:  Have the authors checked other MAPK pathways that have been involved in A71 replication, such as the JNK pathway. Vemurafenib down-regulates JNK and p38, so it is possible that the observed effect may be due to these cascades at least in part.

Response 2: Thank you very much for raising out this important issue. MAPK kinase cascades are complex cellular signaling pathway and involved with hundreds of proteins. As mentioned in the second paragraph of discussion, we believe that activation of the Raf-MEK-ERK pathway inhibits EV-A71 replication organelle (RO) formation. In this stage, we do not exclude the possibility of other modulators of this pathway to play a role in EV-A71 replication cycle. We agree with this reviewer and have added following sentences in the discussion: “Nevertheless, vemurafenib inhibits activation of JNK and p38 kinase signaling cascades, the other two MPAK signaling pathway [12], by influenza A virus infection or tumor necrosis factor alpha (TNFα) treatment [33]. EV-A71 infection leads to activation of Jun-N-terminal kinase 1/2 (JNK1/2) [36] and inductions of the proinflammatory cy-tokines regulated by the MAPK p38 signaling cascades [37], thus regulating cell apoptosis and inflammation. Therefore, vemurafenib probably also limits the activation of the two MAPK signaling cascades induced by EVA71 and thus potentially contributing to its inhibition of EV-A71 infection.”

Round 2

Reviewer 2 Report

I am satisfied with the authors' responses and added supplemental data. 

Reviewer 3 Report

I am happy with the changes carried out by the authors.